# Liver Organoids as an In Vitro Model to Study Primary Liver Cancer

**DOI:** 10.3390/ijms24054529

**Published:** 2023-02-25

**Authors:** Silvia De Siervi, Cristian Turato

**Affiliations:** Unit of Immunology and General Pathology, Department of Molecular Medicine, University of Pavia, 27100 Pavia, Italy

**Keywords:** liver organoid, primary liver cancer, in vitro models

## Abstract

Primary liver cancers (PLC), including hepatocellular carcinoma (HCC) and cholangiocarcinoma (CCA), are among the leading causes of cancer-related mortality worldwide. Bi-dimensional in vitro models are unable to recapitulate the key features of PLC; consequently, recent advancements in three-dimensional in vitro systems, such as organoids, opened up new avenues for the development of innovative models for studying tumour’s pathological mechanisms. Liver organoids show self-assembly and self-renewal capabilities, retaining essential aspects of their respective in vivo tissue and allowing modelling diseases and personalized treatment development. In this review, we will discuss the current advances in the field of liver organoids focusing on existing development protocols and possible applications in regenerative medicine and drug discovery.

## 1. Introduction

Numerous physiological, metabolic, and regulating processes, including bile secretion, glycogen and fat-soluble vitamin storage, drug detoxification, and the synthesis of plasma proteins and coagulation factors, are carried out by the liver [1]. As a result, pathogenic (genetic or acquired) alterations in liver tissue may have significant effects on an individual’s health.

The basic hepatic structure consists of parenchymal cells, hepatocytes and cholangiocytes, and non-parenchymal cells, such as fibroblasts, stellate cells, Kupffer cells, and endothelial cells. In particular, hepatocytes, which are organized in lobules, account for more than half of total liver mass [2], while cholangiocytes are epithelial cells that line the bile ducts and the peribiliary glands and play an important role in the transport of bile constituents from the liver to the duodenum [3].

Primary liver cancers (PLC) are tumours that develop directly in the organ rather than as a result of metastasis [4] and include hepatocellular carcinoma (HCC), intrahepatic cholangiocarcinoma (iCCA), and combined hepatocellular-cholangiocarcinoma (CHC), a rare malignant neoplasm that shows features of both hepatocarcinoma and cholangiocarcinoma [5,6] (Figure 1).

HCC, caused by a malignant transformation of the hepatocytes, accounts for about 85–90% of PLC cases and is one of the most common causes of cancer-related mortality worldwide [6,10]. The remaining 10–15% is represented by iCCA, which is less common than HCC and is caused by intrahepatic biliary tree epithelial alterations [6]. According to the site of origin, in addition to the aforementioned iCCA, CCA also includes a second form, the extrahepatic cholangiocarcinoma (eCCA), which develops outside liver parenchyma and is further classified as perilear cholangiocarcinoma (pCCA), accounting for 50% of cases, and distal cholangiocarcinoma (dCCA), observed in 30–40% of total CCA [11].

Infections with hepatitis B (HBV) and C (HCV) viruses, alcohol abuse (alcoholic liver disease, ALD), metabolic syndrome, obesity, type 2 diabetes (non-alcoholic fatty liver disease, NAFLD), and genetic or immune changes are among the main risk factors for the development of both HCC and CCA [12]. Other established and proven causes that contribute to the development of CCA are biliary tract diseases with resulting chronic infection, such as primary sclerosing cholangitis, cysts of the biliary duct, and parasitic infestations caused by trematodes [13].

The majority of patients receives diagnoses at an advanced stage of the disease, where there are limited and frequently inefficient treatment options, which contributes to the high mortality rate attributable to PLC [14]. Despite efforts, currently, there are no available treatments, and a large portion of the drugs tested over the past ten years are ineffective, failing to pass phase III of clinical trials [15]. The multikinase inhibitor *Sorafenib* [16] and the recently authorised *Lenvatinib* [17] are used as first-line therapeutic choices for HCC targeted therapy, while the only traditional first-line treatment option for patients with CCA at advanced stages of the disease is the combination of gemcitabine and cisplatin; otherwise, the use of folinic acid, fluorouracil, and oxaliplatin (FOLFOX) is used as CCA second-line treatment [18]. However, because of the limited efficacy of these options, there is an urgent need for new therapeutic strategies for PLC treatment.

One of the most significant issues in the preclinical development of regenerative therapies is the lack of appropriate model-based systems that maintain the tumour’s morphologic and functional characteristics, such as three-dimensional architecture, cellular heterogeneity, and cell-cell interactions [19]. In this regard, reliable in vitro models are necessary to increase the knowledge of the molecular and cellular mechanisms underlying PLC progression and provide high-throughput experimental techniques to define biological processes and the efficacy of treatments [20]. As a result, in recent years, the limited clinical value of cell line translation has encouraged researchers to investigate other innovative models for PLC in vitro research. A few in vitro liver models that accurately mimic a working in vivo liver have been developed [21]. In this review, we aim to discuss the recent advances in the field of in vitro liver models with a major focus on liver organoids, a three-dimensional representation of the liver that exhibits accurate micro-anatomy and self-renewal capabilities [22]. In particular, we will analyse the potential innovative applications of liver organoids as a promising new tool for the study of the complexity of liver diseases and the discovery of novel therapies. We will also provide a detailed overview of current protocols and discuss potential novel approaches to address some of their limitations.

## 2. Traditional In Vitro Model to Study Liver Cancer

Over the years, research has led to a greater understanding of crucial physiologic and pathological aspects of liver diseases. Overall, rodent models properly identified less than 50% of the therapeutic response and toxicity of clinically utilised drugs [22]. Therefore, in vitro human cell cultures are the most popular model for studying biological aspects of tumours [14,20], as well as pharmacological mechanisms, efficiency, and toxicity [22].

In the past and still today in vitro studies are based on the use of bi-dimensional cell lines (2D) derived from hepatoma and hepatocarcinoma, as well as 2D primary cultures, providing a useful tool for studying and characterizing molecular events at the base of disease onset and progression, and for obtaining information on the efficacy of treatments [23]. In particular, HepG2, a cell line derived from a liver biopsy of a Caucasian adolescent, is one of the most frequently employed preclinical experimental models for HCC research [14]. HepG2 exhibits typical hallmarks of a hepatic lesion, such as an increased α-fetoprotein (AFP) expression, and expresses distinct hepatic cell functions, such as glycogen synthesis, plasmatic protein and biliary acid synthesis, and cholesterol and triglycerides metabolization. Other cell lines commonly used in HCC research include HepaRG, which originated from a female with HCC, chronic HCV, and cirrhosis, and HuH-7, which are both viable models for studying drug metabolism and carcinogenesis [14].

On the other hand, in the last 40 years, more than fifty cell lines for CCA knowledge have been established [20]. The majority of preclinical research on CCA has been principally conducted in human eCCA cell lines, EGI-1 and TFK-1, and iCCA cells, RBE and HuCC-T1, derived from malignant ascites [20], all of which are representative of a single CCA subtype and thus insufficient for a comprehensive study of its molecular biology [13].

Although 2D cultures are still useful tools for biomarker discovery and drug screening, they have some significant limitations. At first, these cell lines grow in adhesion on a rigid surface with an elongated shape, creating a monolayer where interactions only occur between adjacent cells, and typical functions, such as signalling, proliferation, and migration, are altered [24]. Moreover, 2D cell cultures, which can only develop in two dimensions, have a higher proliferative capacity compared to in vivo conditions and are exposed to uniform concentrations of cell medium nutrients [25]. In addition, compared to patient-derived tissue, gene expression analysis of immortalized cell cultures revealed a significantly limited sensitivity to drug treatment that can easily induce apoptosis. This frequently results in incorrectly promising several molecules that, when tested in vivo, fail to provide the desired results [24].

As previously mentioned, another crucial model for research on PLC is the primary 2D human cell cultures, directly derived from cancer patients’ tissue samples, which were developed to overcome some limitations of the conventional cell lines [19]. Due to their ability to retain representative hepatocytes characteristics, such as expression levels of metabolizing enzymes and liver-specific markers, primary cultures represent a more reliable tool for in vitro research on hepatic metabolism, drug toxicity, and viral infections liver-related [26]. However, primary hepatocytes have a limited lifespan in culture, lasting only a few days, leading to a decrease in hepatic function in vitro [26] and necessitating the expensive donation of fresh material [19]. Additionally, the process of derivation of primary cultures is laborious since it is possible to detect an unwelcome increase in healthy cell fractions that must be eradicated [20]. Despite several advantages, such as easy reproducibility and cheaper costs, 2D cell techniques remain a too simplified model of tumour tissue, which is, instead, excessively heterogeneous and characterized by a complex and dynamic microenvironment [27].

## 3. Three-Dimensional Cell Culture (3D)

In recent years, research has focused on developing three-dimensional (3D) cell models that may be derived from both patient biopsies and commercially available 2D cell lines described above. As shown in Table 1, in comparison to conventional 2D cell cultures, 3D systems provide a more accurate preservation of the in vivo conditions, processes, and microenvironment in which the tumour arises and develops [24], allowing the evaluation of several biological aspects, including proliferation, morphology, and cell-cell and cell-microenvironment interactions [14].

### 3.1. Spheroids

One of the first discovered 3D systems is represented by spheroid, a three-dimensional cellular aggregate with a spherical shape enriched in stem-like cell population but with too low complexity to mimic tumour organization [27]. Spheroids can be produced from primary cultures or cell lines that have been cultured as single or multi-cell suspensions [28]. To enable the development of floating spheres, the single-cell suspension is typically maintained in the absence of a matrix, in ultra-low attachment plates [29], and in serum-free conditions [28]. The use of spheroid is extensive and includes drug screening, immune interaction modelling [30], and the possibility of setting up co-culture systems with both healthy and cancerous cells, which aims to implement the understanding of angiogenesis and tumour metastatic mechanisms.

Wang et al. developed efficient and reproducible agarose hydrogel microwells to produce uniform-sized multi-cellular tumour spheroids, which offer better mimicry of traditional solid tumours and allow the evaluation of some anti-cancer drug candidates’ effects, starting from cells of HCC-patients with abnormally high expression of fibroblast growth factor receptor 4 (FGFR4) [31]. In another study, liver spheroids were established from iCCA cell lines HuCC-T1, CCLP1, and CCA4 and then characterized, revealing an increased expression of key genes involved in self-renewal, drug resistance and survival, as well as stem-like surface markers [32].

### 3.2. Scaffold-Based 3D Systems

Another viable 3D cell culture method is represented by scaffold-based systems, which embedded cells into a physical matrix, allowing them to aggregate, proliferate, and migrate [14]. Scaffolds are made up of a multitude of materials with varying porosity, permeability, and mechanical stability, to replicate the microenvironment of the extracellular matrix (ECM) of tissues and tumours [33]. Among different existent scaffolds, the distinctive hydrogels can mimic the characteristics of the ECM, allowing soluble factors like cytokines and growth factors to pass through the gel tissue-like support [34]. Hydrogels are incredibly adaptable since their preparation could differ depending on the experiment being conducted. There are both natural hydrogels that are typically made from natural polymers such as fibrinogen, collagen, hyaluronic acid, gelatin, and alginates, and synthetic hydrogel, made with polymeric materials with chemically defined bases, such as polyethylene glycol (PEG), polylactate (PLA), or polyvinyl alcohol (PVA) [24]. A natural hydrogel widely used in 3D cell culture is Matrigel. This is derived from secretions of the Engelbreth-Holm-Swarm murine sarcoma and appears as a soluble material rich in collagen IV, laminin, proteoglycans, soluble heparan, and entactin that can solidify at 37°C and mimic the properties of the base membrane matrix [35].

Recently, Turtoi et al. aimed to create a new 3D cell model of HCC, seeding HepG2 cells in a hyaluronic acid-based scaffold, in order to evaluate the cytotoxicity and apoptotic response to the anti-tumour agent cisplatin [36]. They demonstrated that the hyaluronic acid-based system allowed cells to proliferate into larger aggregates, showing liver-like functions, expressing main hepatocyte-specific biomarkers, such as albumin, bile acids, transaminases, and sensitizing the hepatocytes to the anti-tumour effect of cisplatin [36]. It also fabricated scaffolds for 3D culture models of CCA, using a CCA cell line (KKU-213A), by combining silk fibroin with hyaluronic acid, heparin sulfate, and gelatin, which could yield cancer stem cells and more accurately mimic tumour behaviour better than 2D systems, in terms of cell proliferation, microenvironment representation, and drug sensitivity [37].

### 3.3. 3D-Bioprinting and Organs-on-a-Chip

Among other in vitro 3D models, there are 3D bioprinting, and organs-on-a-chip, which are both technologies derived from the combination of cell biology with engineering and biomaterials technology [14].

Cell models created with 3D bioprinting are innovative platforms based on the use of bioinks containing living cells, decellularized ECM constituents, nutrients, growth factors, and biomaterials with the purpose of engineering 3D constructs with tissue-like architecture [38,39]. As a result, bioprinting technology may create systems that successfully replicate the ECM, improving cellular proliferation rates and responses to chemotherapeutic drugs compared to conventional 2D models [40].

In a recent research, authors developed a 3D model with HepG2 cells, using 3D-bioprinting technology, in order to demonstrate the different effects and pharmacodynamics of some anti-tumour drugs between 2D and 3D HepG2-derived systems [41]. Moreover, Xie et al. proved that 3D bioprinted models are capable of performing drug screening through the establishment of patient-derived HCC hepatorganoids [42]. Current advances in 3D bioprinting technology have motivated bioengineers and scientists to also create methods for “printing” in vitro tumour-mimicking models in order to study the molecular mechanisms behind tumour growth. An example is represented by the bioprinter platform made by Li et al., which includes in a single system both RBE (an iCCA cell line) and stromal cells, including human umbilical vein endothelial cells (HUVEC), fibroblasts (CCC-HPF-1) and human monocyte leukaemia THP-1, demonstrating how stromal cells affected the proliferation, invasion, stemness, and drug resistance of CCA cells. As a result, this 3D bioprinted CCA model could be employed to more accurately mimic the tumour microenvironment, potentially serve as a robust, clinically accurate platform for preclinical research and drug testing, and offer a viable substitute for animal models [43].

Moreover, organ-on-a-chip models simulate real synthetic microenvironments, integrating living cells that can mimic the in vitro functions of an organ. New studies have successfully replicated the connection of several organ-on-a-chips to create body-on-a-chip models that represent multi-organ interactions and study the metastasis process in cancer in a more thorough manner [27]. In addition, in order to investigate the dynamic evolution of the tumour through proliferation, angiogenesis, and intravasation processes, vascularized tumour-on-a-chip models were created [44]. In contrast to traditional 2D models and animal testing, liver-on-a-chip technologies enable more effective management of the cellular microenvironment, increasing hepatocytes activity, simulating cellular responses to medicines in vivo, and more closely simulating liver physiology [45]. In one study, induced pluripotent stem cells were used to reconstruct the liver acinus, including its vascularized form, in conjunction with the pancreas and adipose tissue; additionally, fluorescent protein biosensors were added to the device to assess insulin resistance and the production of reactive oxygen species [46]. Thanks to this system, authors can investigate liver-specific biomarkers, identifying the progression from NAFLD to steatohepatitis within an experimental timeline [47].

### 3.4. Organoids

Organoids are an in vitro 3D model that recapitulates some structures and functions of the corresponding in vivo organ, not visible in 2D cultures, derived directly from the dissociation of specialized epithelial tissues, from embryonic stem cells (ESCs) or induced pluripotent stem cells (iPSCs), all capable of self-renewal and self-organization [48].

It has been discovered that organoids are a powerful system for studying development and regenerative processes as well as for understanding some diseases [49]. These models also provide new tools for translational research, making them a promise for drug development and personalized treatments [50]. Organoids offer the following several benefits: they combine the tractability of in vitro cell cultures with the architecture and differentiation of in vivo models, making them comparable to standard 2D cell lines in terms of long-term culturing, cryopreservation, and genetic manipulation [51].

Unfortunately, some significant restrictions on the use of organoids have been described, most of which are related to laborious protocols; for example, the development of tumour-specific organoids has only been successful in patients with highly differentiated tumours with high proliferative rates, ruling out the possibility of using patients who are still in the early stages of their disease [23]. Furthermore, because cancer is characterized by a heterogeneous TME, in which both cellular (epithelial cells, fibroblasts, stem cells, endothelial, and immune cells) and non-cellular (ECM, cytokines, chemokines, and growth factors) components are essential for the development and progression of the tumour, the lack of all of these components in a single 3D system represents a significant limitation [27,52]. However, despite the lack of reliable experimental protocols and the high cost of implementation, these 3D systems provide innovative tools for understanding the mechanisms underlying tumour progression.

Furthermore, thanks to their ability to show high levels of genomic stability and mimic the heterogeneity observed in real tumours, organoids can be propagated for long periods with few genetic variations. Another aspect is the employment of organoids may decrease the requirement for using animal models and, thus, any associated animal ethics issues [53].

The first experiments that enabled the development of organoids were based on the isolation, from murine intestinal epithelium, of single leucine-rich repeat-containing G-protein-coupled receptors 5 (LGR5) positive adult stem cell, capable of self-renewing. These cells LGR5^+^ were placed in suspension, embedded in Matrigel with a medium containing a variety of growth factors, in order to mimic the combination of signals that persist in the niche, giving rise to three-dimensional structures with a total cytoarchitecture that is similar to that observed in vivo [54].

Following these unexpected extraordinary results, in recent years, organoids derived from various types of tumours have been described, including the brain [55], prostate [56], pancreas [57], colorectal [58], breast [59], bladder [60], and liver cancer [23,61,62], starting to embryonic stem cells (ESCs), induced pluripotent stem cells (iPSCs) and adult stem cells (ASCs) [27].

## 4. Liver Organoids

Huch and colleagues have reported the discovery of the first system of intestinal organoids obtained from epithelial biliary LGR5^+^ cells that were isolated from hepatic injury mice models and placed in a cultured medium enriched with R-Spondin 1 (R-Spo1) and Wnt3a, both WNT pathway activators [61].

The organoids thus obtained, termed cholangiocyte-derived organoids (chol-orgs), are an accurate in vitro model that captures the main characteristics of the biliary epithelium in vivo in terms of morphology, functions, and markers expression. However, they also exhibit higher levels of foetal markers and lower levels of mature markers, indicating a partial differentiation of the cholangiocytes [19]. In contrast to chol-orgs, recently it is developed hepatocytes-derived organoids (hep-orgs), organoids derived from primary hepatocytes, which exhibit phenotypic properties of hepatocytes more accurately in terms of molecular expressions of particular markers, as well as functional characteristics [63]. Using an appropriate differentiation medium that includes some new factors, such as fibroblast growth factor-19 (FGF-19), DAPT (a Notch inhibitor), and dexamethasone, chol-orgs at early passages may be differentiated into cells with a hepatocyte-like phenotype that are able to secrete albumin and carry out a variety of hepatic functions [19,53].

As a demonstration of the hepatoblasts’ bipotential plasticity, from a single subpopulation of LGR5^+^ cells, both chol-orgs and hep-orgs can be produced [25,64].

Moreover, the culture environment, both in terms of signalling and cell type, has a crucial role in the development and maintenance of organoids [65]. The organoids resulting from the hepatic tumour may be established by adult tissues surgically exported [62] or, more recently, by needle biopsies of patients affected by HCC, CCA, and CHC [23]. ESCs and iPSCs are alternative sources for the in vitro generation of organoid models [66].

During organoid formation, the starting cell population begins to assemble in a specific signalling environment, where it is necessary to provide signals related to liver development in order to trigger self-organization [25,64] (Figure 2).

Human liver organoids from adult tissues need the identification of mitogenic signals through a variety of factors, including epithelial growth factor (EGF), fibroblast growth factor (FGF), and hepatocytes growth factor (HGF) [61,67,68]. Forskolin (FSK), an activator of cyclic adenosine monophosphate (cAMP) and the inhibitor of TGF-β signalling, A8301, are also added to the culture medium to allow long-term expansion [61,67]. A few days after seeding, ROCKi, an inhibitor of the Rho-associated kinase protein (ROCK), is added to the medium to prevent the apoptotic process [61,69] (Figure 2).

In addition, as described by Peng et al. approach, liver-resident macrophages release large amounts of inflammatory cytokines, including TNF-α, following liver damage to help in regeneration; based on these findings, hepatocytes growth was certainly aided by the addition of 100 ng/mL TNF-α to the hep-orgs culture medium [70].

According to the protocols from Huch [61], Broutier [71], and Nuciforo [23] laboratories, all factors, with their respective concentrations, added to the culture medium for liver organoids development are illustrated in Table 2.

For human organoids generation from iPSCs, changes in the culture medium were applied, with the WNT signalling inhibition [53], and the addition of different nutrients, such as activin A, bone morphogenic protein 4 (BMP4), and phosphoinositide 3 kinase inhibitor (PI3Ki) that help the differentiation of iPSCs through stages, resembling human liver during its embryonic development [66]. To differentiate the hepatic progenitors into hepatocytes, HGF and Oncostatin M were also added in the medium [72].

To allow three-dimensional suspension growth, it is necessary to provide the organoids with structural support using hydrogels such as Matrigel or Cultrex Basement Membrane Extract (BME) [19] (Figure 2).

At the level of Disse space, hepatocytes are located near the ECM, linked to collagen type I, fibronectin, and laminin, affecting cell proliferation, differentiation, and migration. In particular, biochemical signals, such as the composition of the matrix, and mechanical properties, such as rigidity, act on the differentiation of the liver progenitor cells toward the hepatocytes or cholangiocytes lines [73]. For this reason, it is crucial to replicate both the biochemistry and the biomechanics of the native ECM of the in vitro liver tissue.

As mentioned above, Matrigel has an advantageous protective complexity that enables it to mimic the structure of basal membrane; on the other hand, its murine origin has an elaborate process that results in elevated batch-to-batch variations in terms of composition and rigidity that interfere in vivo applications [73]. Recently, it has been discussed new approaches that could replace the use of Matrigel with alternative biological hydrogels that are appropriate from a chemical and physical standpoint in the regulation of mechanical properties [74]. Based on these findings, it is possible to intervene by altering the component ratio of miscellaneous components or by reinforcing sticky gels with more stable mechanical and spatial structures [73]. This is especially significant when used in the organoids culture since they are significantly controlled through mechanotransduction [75]. Synthetic hydrogels’ use is also becoming more successful, but since synthetic polymers lack biological activity, ECM’s biological functions must be restored by including biomolecules.

Decellularized ECM acquired from both human and animal donors has also been used to develop some organoids accurately recapitulating the composition, structure, and vascularization of native ECM. The particular ECM for the liver may be obtained from a portion of surgical resection of a patient’s damaged liver or unsuitable livers for transplantation [33]. Recently, Willemse et al. described the culture and the expansion of human cholangiocyte organoids in hydrogel derived from decellularized liver tissue, showing the preservation of the cholangiocyte-like phenotype and the expression of selected cholangiocyte markers [76].

Different available materials to mimic the ECM in the generation of liver organoids are listed in Table 3.

According to Nuciforo and Heim, the success rate of the experiment varies significantly between the generation of chol-orgs and hep-orgs: one-fourth of all cholangiocytes can start a transformation into an organoid with extremely rapid proliferation and long in vitro expansion, while just one hepatocyte out of every 100 produces hep-orgs, which proliferates more slowly and divides every 50–75 days when derived from the adult liver [19].

Once obtained, it is possible to cryopreserve liver organoids for long-time periods that can reach as long as 1–2 years, allowing the creation of biobanks of heterogenous tumour organoids, in which each sample is representative and exhibits a variety of histopathologic and molecular PLC characteristics [23].

## 5. Liver organoids Characterization

Following generation, PLC-derived organoids could be characterized both at the molecular level using whole genome sequencing or RNA sequencing in order to detect gene expression or compare the presence and maintenance of some mutations, and proteomic techniques, such as immunohistochemistry and immunofluorescence that enable the assessment of the potential presence/absence and the quantification of specific markers levels (Figure 2).

The transmembrane glycoprotein epithelial cell adhesion molecule (EpCAM) is one of the markers that has received the most attention for characterizing CCA-derived organoids. In the liver, EpCAM is a biliary marker, often not detected in mature hepatocytes [77], which has a physiological role in mediating intercellular adhesion in epithelial tissues and occurs at an early stage of the neoplastic transformation of CCA cells [78]. Another potential biomarker for CCA is Sex Determining Region Y-box 2 (SOX2), a transcriptional regulator in maintaining regeneration for embryonic stem cells. Numerous malignancies depend on SOX2 for carcinogenesis and tumour growth, and in CCA, SOX2 over-expression was linked to poor overall survival, increased cell proliferation and invasion, and reduced cell apoptosis; however, its exact role in CCA must be clarified with more studies [78].

Other two PLC biomarkers that have been studied include cytokeratins 7 and 19 (CK7 and CK19), which are crucial for maintaining epithelial barriers, regulating innate immunity, and cell adhesion, proliferation, and differentiation [79]. It has been observed that these two molecules are useful histochemical markers for the differential diagnosis of HCC and iCCA [80], as well as potential post-operative prognostic factors for CCA [81]. Therefore, a greater sense of security regarding the true nature of cells cultured is provided by the presence of these molecules in tumour organoids.

On the other hand, the most significant HCC markers are albumin (ALB), hepatocyte nuclear factor 4 (HNF4), and α- fetoprotein (AFP) [62]. This latter one represents a marker of liver function, such as synthesis and secretion, typical of differentiated hepatocytes [62], and its up-regulation is present in more than 40% of tumour samples [82]. Moreover, the panel of immunohistochemical markers composed of heat shock proteins 70 (HSP70), glypican-3 (GPC3), and glutamine synthetase (GS) was recommended for the differentiation of early HCC. In particular, the HSP70s family was revealed to have a critical role in the development and progression of various cancers, including HCC [83]. HSP70s are involved in protein synthesis and transport, in order to maintain protein homeostasis, and it was observed that an over-expression of several HSP70s in HCC is associated with the overall survival, tumour grade and cancer stage [84]. In addition, GPC3 is considered a potential early diagnostic marker, associated with poor prognosis, of HCC, due to its involvement in cell proliferation through WNT/β-catenin pathway activation [82]. Recently studies evidenced how GPC3 could be a potential drug target that has significantly reduced tumour growth and prolonged survival in Phase I clinical trials [85]. Finally, GS levels also gradually increase with the development of HCC and were observed in its involvement in promoting epithelial-to-mesenchimal transition (EMT) [86].

During the last decades, the presence of the nuclear antigen Ki-67, a marker of tumour cell proliferation capability, has also received considerable attention [87]. This protein undergoes a rapid degradation during the G1 phase of the cell cycle, causing a reduction in intracellular levels in cells that are quiescent or have limited proliferation [88] and an increment in tumour cells that have a rapid division [89], underlying a correlation between Ki-67, the severity of the tumour, and the likelihood of a favourable prognosis [90].

## 6. Liver Organoids Potential Applications

As a result of the ability to use liver tissue samples for the assessment of organoid cultures, research is moving toward the use of these 3D systems as disease models, in addition to being an extremely helpful tool for precision and personalized medicine [74] (Figure 3).

Recent studies have shown that tumour-derived organoids are capable of retaining the morphological characteristics and biomarkers of the original tumour tissue while also preserving the patient-specific gene expression profile, even when cultivated for extended periods [23,62]. Using gene editing techniques, such as CRISPR/Cas9, it is thus possible to engineer organoids, introducing or correcting certain mutations that may be appropriately studied and assessed for their function and pathogenicity [91,92].

Additionally, thanks to their peculiar metabolic capacity, liver organoids are promising tools for the development of new treatments for clinical use. Indeed, due to their ability to be expanded in vitro for long periods and to be cryopreserved, biobanks have been developed to be used as platforms for high-throughput drug screening of anti-cancer treatments [19,93]. Biobanks of healthy organoids, on the other hand, can represent a useful predictive investigation tool for the in vivo toxicity of drugs [25].

As previously described, one of the main issues with organoids is related to the fact that these systems are characterized by a single cellular type of representative of the neoplastic epithelium and do not fully represent the typical multi-cellular tumour environment. One of the solutions is represented by the setting up co-culture systems of liver tumour organoids with a variety of cell types, including patient-derived immune cells or cancer-associated fibroblasts (CAFs), thus offering a promising tool for modelling the dynamic interactions between expanding cancer cells and the immune system [30,66].

Recent improvements in co-culture techniques make it possible to create ever-more complex and cutting-edge systems, such as vascularised liver organoids, and to research host-pathogen interactions in vitro, such as the host-HBV/HCV interactions, a key factor in the development of PLCs [94,95]. An example is represented by Natarajan et al. who developed a co-culture system to study adaptive immune responses to HCV, using patient-derived CD8^+^ T-cells specific for HCV non-structural protein 3 to generate liver organoids [96].

During recent years a technique known as “interface liquid-air” (ALI) has also been developed, allowing the combination of organoids with both epithelial and stromal cells using standard Boyden chambers. The functioning of this system is based on cells that are embedded in ECM gel and placed on the upper surface of cell inserts with a below porous membrane, directly exposed to oxygen, while nutrients and growth factors are supplied from the external medium by diffusion through the porous membrane on the lower surface [27].

Liver engineering organoids may be further used in the future to study the early stages of liver tumours, offering an innovative perspective on preventive therapy. The advantages of maintaining the molecular and structural abnormalities brought about by oncogenes make organoids an ideal in vitro model for understanding oncogenic processes during tumour development [1].

In recent years, additional advancements in the organoid model have resulted in the creation of the organoid-on-a-chip, a micro-fabricated, integrated system that combines the architectural and genomic recapitulation of organoids with the highly customised flexibility of organ-on-a-chip models [97]. Numerous issues with traditional organoid models are resolved by the organoid-on-a-chip, such as a major control over the organoids’ microenvironment. Moreover, the organoid-on-a-chip model may also contain vascular and immunological components, significantly enhancing its therapeutic relevance in drug screening and clinical trials. A vascularized cancer model is required for researching tumorigenesis and metastasis because abnormal angiogenesis is a key component of carcinogenesis [53].

Another interesting area is the possibility of using liver organoids as instruments to simulate significant chronic liver diseases, such as NAFLD and liver fibrosis [98]. Growing evidence indicates that NAFLD is becoming a dominant cause of HCC [99] and CCA [100], but the mechanisms of NAFLD progression are largely unknown. NAFLD is characterized by intracellular deposition of lipids in hepatocytes, often associated with a wide spectrum of metabolic abnormalities, such as dyslipidemia, hypertension, and insulin resistance. The disease then ranges to non-alcoholic steatohepatitis (NASH), a more severe condition that includes inflammation and additional hepatocyte damage and can progress to cirrhosis [101]. For example, by exposing liver organoids to free fatty acids (FFAs) in perfused 3D cultures over an extended length of time, it is enabled to define the pathological characteristics of NAFLD. In this way, liver organoids could show lipid droplet production and triglyceride buildup after FFAs induction, demonstrating increased expressions of genes linked to lipid metabolism and highlighting the aberrant lipid metabolic pathway in NAFLD [102].

In conclusion, a significant characteristic of tumour organoids is the ability to predict their potential for in vivo metastasis, in addition to maintaining the genetic model of the primary tissue [74]. The animal models receiving transplants of liver organoids have shown encouraging outcomes [63,69,70]. However, the protocols must be further improved, to increase the rate of engraftment and to promove circulation in patients for the delivery of oxygen and nutrients [103]. The final stage in making tissue engineering a reality for the treatment of liver disease is to find solutions to these problems [74].

## 7. Conclusions and Future Directions

Due to the lack of reliable in vitro models and available treatments for PLC, there is an urgent need for an improved preclinical tumour system that can mimic the genetic background and architecture of the primary tissue. Moreover, significant variations between human and mouse physiology, metabolism, size, and longevity are among the shortcomings of in vivo animal models [104].

The 3D organoid systems represent an enormous promise for solving these limitations, besides providing several practical applications that potentially change biomedical research, drug development, and disease modelling. Traditional 3D cultures have faced issues in order to accomplish the right control of organoid production and to realise the complex microenvironment of a specific organ due to the quick development and broad needs of organoid technology [1]. Until now, the use of organoid models has permitted the development of novel possible treatments as well as a better knowledge of the underlying mechanisms of disease onset and progression.

For efficient diagnosis and therapy decisions, patient-derived organoids represent an innovative option, thanks to their strong advantage of retaining personalized genetic information [105]. Moreover, the creation of liver organoids by bioengineering has the potential to produce more physiologically realistic and biomedical useful specimens. The potential to reproduce in vitro liver epithelial cells has been improved by the ability of liver cells to produce liver organoids.

Despite the fact that liver organoids are among the most advanced human cell-based 3D liver models, and organoid-based drug testing may accurately predict clinical outcomes in personalized medicine and drug toxicity and efficacy evaluation [105], there are still several issues that need to be addressed, such as increased costs, absence of highly reproducible results, lack of other TME cell types and 3D culture platforms to model their interactions, and use of animal-derived 3D-matrices [106]. In part, these limitations can be attributed to the current use of non-standardised and well-defined protocols, which introduces technical variability into in vitro organoid cultures and reduces their accurate representation of cancer’s intrinsic biological heterogeneity [106].

Recent advancements in microfabrication techniques offer the ability to standardise cancer organoid derivation, analysing how the size of the starting cell cluster affects the rate of organoid development, for example. These improvements in cancer modelling will be well complemented by the increased availability of methods that monitor and measure organoid proliferation at the cellular level [106]. In addition, creating multi-cellular liver organoids in which epithelial cells interact with endothelial, mesenchymal, and immunological cells is necessary for the disease modelling of PLC, where the microenvironment plays a crucial role [25]. Microphysiological systems represent a promising approach for building organoid/tumour-on-a-chip models with more tissue complexity, including the incorporation of mature vasculature [107]. Several microfluidic devices have been developed to simulate how cancer interacts with vascular networks, allowing the evaluation of cancer extravasation, drug delivery, and tumour growth [107]. Finally, the implementation of engineered matrices animal-free, using hyaluronic acid or PEG, for example [108], will represent a future opportunity for high batch-to-batch reproducibility, standardisation of organoid development and culture protocols, and for understanding the roles of the ECM in regulating patient-specific tumours.

Because of the potential applications of these 3D models, in the future, organoids will open the road to the regeneration of injured or diseased organs, a proposal that was previously thought to be unlikely to be accomplished in medicine. The ability of liver organoids to regenerate diseased livers may be very promising, and for this reason, the goal of current research is the creation of organoid liver buds that can be delivered to patients who are in urgent need of a liver transplant via the portal vein [53]. In this way, a structured patient-based treatment system may require everyone to have organoid tissue maintained in large-scale biobanks in the future, improving the core strategies and tenets of personalized medicine. Furthermore, working closely with bioengineers to add blood vessels to liver organoids may be considered crucial, and doing so is a feasible solution to the problem of the limited nutrition availability that eventually affects the development of organoids [25].

In conclusion, the repeatability of organoid systems, the addition of cells from different functional lineages, and the use of gene editing techniques for the acquisition of complex organoids, therefore, opened up new research fields.

## Figures and Tables

**Figure 1 ijms-24-04529-f001:**
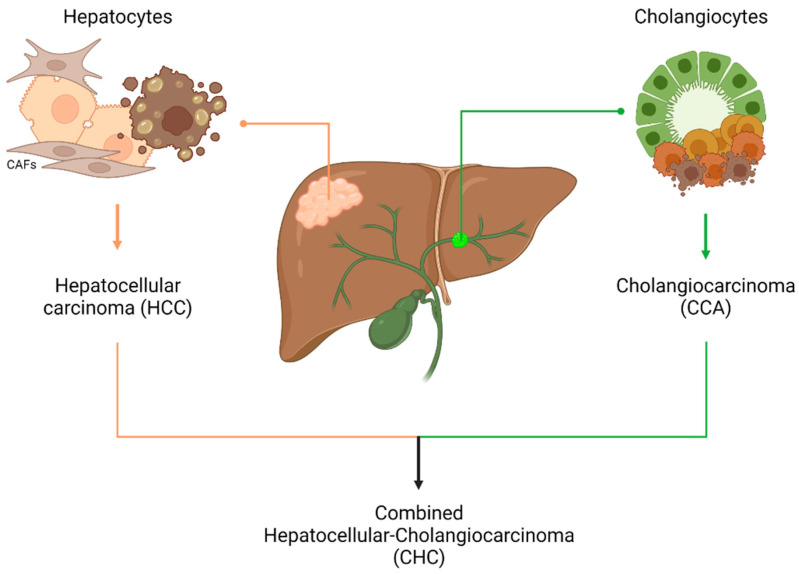
Primary liver cancers, including HCC (85–90%) and iCCA (10–15%) [6], are resulting from the malignant transformation of hepatocytes and cholangiocytes [7], epithelial cells that make the liver parenchyma. A third form is represented by CHC (0.4–14.2%) [8,9], which combine features of both HCC and CCA.

**Figure 2 ijms-24-04529-f002:**
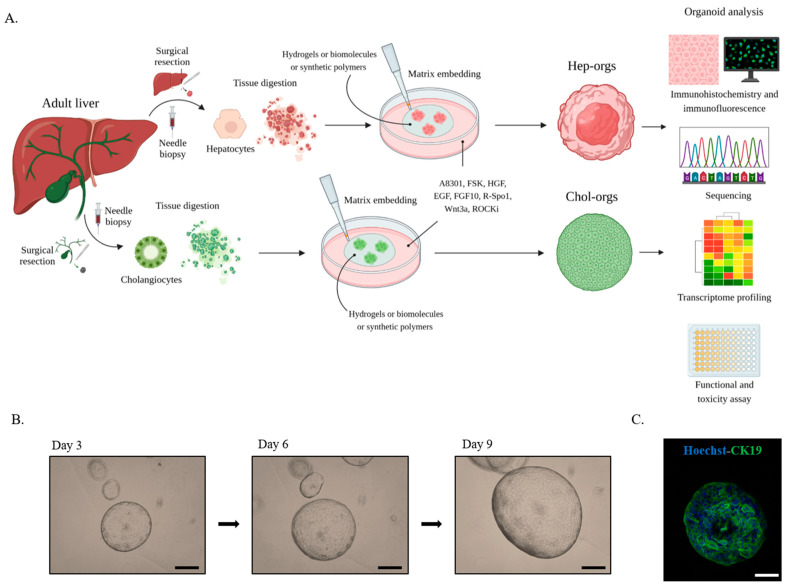
(**A**). Representation of the generation of liver organoids from ASCs. Under sterile conditions, biopsies undergo mechanical and enzymatic digestion to isolate liver cells for organoid development. Cells are then seeded in a suitable matrix that mimics the ECM, and after polymerization, the culture medium containing a cocktail of growth factors is added. For organoids’ characterization several techniques, including immunofluorescence and sequencing, could be used. (**B**). Representative bright-field images of tumour organoids from a iCCA patient. Organoids were imaged every three days, growing like a cystic structure. Scale bar: 200 μm. (**C**). Representative immunofluorescence analysis for CCA marker cytokeratin 19 (green) on tumour organoids. Nuclei were counterstained with Hoechst33342 (blue). Scale bar: 50 μm.

**Figure 3 ijms-24-04529-f003:**
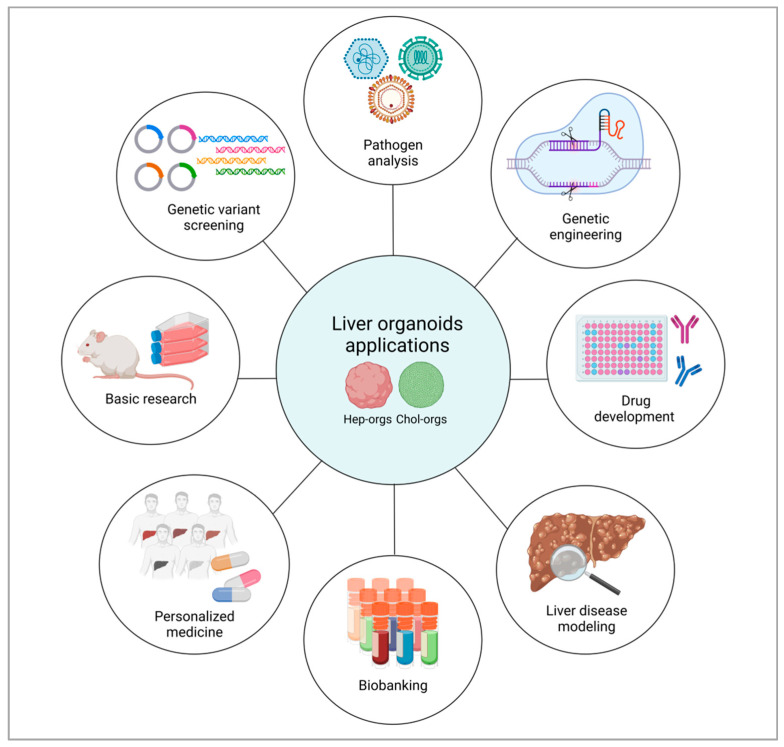
Applications of liver organoids. Liver organoids uses range from basic research to liver disease modelling to personalized medicine that aims the identification of patient-specific responses to drugs.

**Table 1 ijms-24-04529-t001:** Main differences between in vitro 2D and 3D models.

Characteristics	2D Cell Culture	3D Cell Culture
Cell morphology	Flat and elongated morphology	Predisposition to maintaining natural cell shape
Type of interaction	Adjacent cells interactions on a monolayer	Cell-cell and cell-extracellular matrix interactions
Exposure to culture medium substances	Equal exposure to culture medium’s nutrients and growth factors	Exposure to additional medium factors based on gradient
Drug sensitivity	High sensibility, superior to reality	Greater resistance More realistic representation of therapeutic potential
Expression levels	Different expression levels compared to in vivo levels	More accurately identification of in vivo gene expression levels
Use and analysis	High repeatability and easy data interpretation	Difficulty in reproducing experiments and data interpretation
Cost	Low	Expensive

**Table 2 ijms-24-04529-t002:** Components of liver organoids expansion medium. R-Spo1 and Wnt3a were removed from medium of healthy organoids after 3 days of culture [23,61].

Components	Concentrations	Functions
B-27	1:50	Serum-free supplement, without vitamin A, it increases differentiated cell vitality during long term expansion culture condition.
N-2	1:100	Serum-free supplement, it promotes neuronal primary cell cultures’ growth.
Nicotinammide	10 mM	Anti-inflammatory agent, it controls cell metabolism, mitochondria functionality and energy production.
N-acetil-L-cisteine	1.25 mM	Mucolytic agent, with cytoprotective, anti-inflammatory and antioxidant effects, through NF-Kb and HIF-1α regulation and ROS levels modulation.
Forskolin	10 µM	Diterpenes, agonist of cAMP pathway, it has an anti-inflammatory effect and promotes mRNA expression in primary hepatocytes; it supports long-term expansion of organoids.
Y-27632 (ROCKi)	10 µM	Rho-kinase inhibitor, it blocks apoptosis process.
A83-01	5 µM	TGF-β signalling inhibitor, it blocks the epithelial to mesenchimal transition TGF-β induced; it supports long-term expansion of organoids culture.
[Leu^15^]-Gastrin I	10 nM	Essential for digestive system, gastrin stimulates the production of gastric acid from paretial cells and prolong the survival time of liver organoids.
FGF-10	100 ng/mL	Growth factors with mitogen effect, they promote cell proliferation, differentiation, and survival.
EGF	50 ng/mL	
HGF	25 ng/mL	
Noggin	100 ng/mL	Bone morphogenic protein (BMP) inhibitor.
R-Spo1	10%	Agonist of WNT/β-catenin and WNT/PCP pathways and ligand of LGR5^+^ receptor, it improves efficiency of organoids expansion.
Wnt3a	30%	Agonist of WNT pathway, it promotes stem cell LGR5^+^ proliferation, essential for organoids expansion.

**Table 3 ijms-24-04529-t003:** Different types of materials are used to mimic ECM during organoid generation. PEG: polyethylene glycol; PLA: polylactate; PVA: polyvinyl alcohol; PLGA: poly lactic glycolic acid; PCL: polycaprolactone.

Scaffold	Materials	Advantages	Disadvantages
Natural	Matrigel, Cultrex Basement Membrane Extract (BME)	Commercially available; widely used in the majority of developed protocols	Indeterminate culture system with no control over mechanical properties and a lot-to-lot variability; may not include all chemical signals required for differentiation; immunogenicity
	Decellularized tissue	Developed organoids can be large and still retain mechanical qualities and natural chemical signals	Difficult preparation, limited by donors’ resources
	Biomacromolecules (collagen, alginate, hyaluronic acid, silk)	Low cost and wide availability	Lack of retained structural information, absence of the required chemical signals, and lot-to-lot variability
Synthetic	PEG, PLA; PVA PLGA, PCL	Improved control over mechanical and chemical features; easily reproducible experiments; variable degradation rate	It requires the functionalization using peptides that are attached to the cell membrane; potential cytotoxic issues

## Data Availability

More information is available from the corresponding author upon request.

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
