# Peer review of "Liver Organoids as an In Vitro Model to Study Primary Liver Cancer"

_ijms, 2023, doi:10.3390/ijms24054529_

Round 1

Reviewer 1 Report

General Comments:

The manuscript entitled "
Liver organoids as an in vitro model to study primary liver cancer" is comprehensive review of   the current advances in the field of liver organoids focusing on existing development protocols and possible application in regenerative medicine and drug discovery.

My comments regarding this paper from the Title of article is appropriate. Abstract is good arranged. Introduction is good organized with sufficient literature review. Methods are evidence based. Results show findings clearly. Conclusion is logical and appropriate and reflected the title of study. References are related to the issue. I think the figures, tables and highlights are important and help to better understanding of the subject. After reviewing this manuscript, I do have some concerns about the research.

1.     Please site the reference in the sentence “Primary liver cancers (PLC) are tumors that develop directly in the organ rather
than as a result of metastasis and according to starting cells alterations are divided into hepatocellular carcinoma (HCC) and cholangiocarcinoma (CCA)”

2.     Please revised the figure 1. In the figure legend, author elaborate that. “Primary liver cancers, including HCC, CCA and CHC, are resulting from the malignant
transformation of, respectively, hepatocytes and cholangiocytes, epithelial cells that make the liver parenchyma”. In my opinion according to this figure CHC caused by HCC and CCA; it’s mean does not match with the stataement of “Primary liver cancers, including HCC, CCA and CHC, are resulting from the malignant transformation of, respectively, hepatocytes and cholangiocytes” . Please clarify this issue. It will more interesting if author also put the percentage of PLC according to these type of PLC HCC, CCA and CHC.

3.     Please site the reference” HCC, caused by a malignant transformation of the hepatocytes, accounts for about 85-90% of PLC cases and is one of the most common causes of cancer-related mortality worldwide”.

4.     Please mention the type of these tumor types in the statement of “and genetic or immune changes are among the main risk factors
for the development of both of these tumor types.

5.     In Paragraph 6. Author just mention the problem of current situation of lack of treatment for PC. It is better if author clearly explained the gap and solution for the situation to explain why author interested to reviewing the study about “Liver organoids as an in vitro model to study primary liver cancer”.

6.     In the first paragraph of part 2. Traditional in vitro model to study liver cancer; the paragraph it too short

7.     It will more interesting if author can elaborate the in vitro model for specific liver cancer such as HCC, CCA and CHC.

8.     Author should explain the innovations of this study?

9.      Author should explain the "limitations" of the study at "Discussion" section.

10.  Author should explain in detail the recent advancements in three-dimensional in vitro systems , such as organoids and it is the strength for Liver cancer in vitro model.

Reviewer 2 Report

The review article “Liver organoids as an in vitro model to study primary liver cancer” by Siervi and Turato discusses recent findings related to the development and importance of Liver organoids.

It’s an interesting and translationally relevant for liver cancer. The review provides a comprehensive understanding about the organoids as well as discusses pros and cons of different model system. Only one minor comment needs to be improved for the betterment of the manuscript. On page 4, making subsections for section 3 for topics like spheroid, scaffold-based systems, 3D-bioprinting, and organs-on-a-chip will improve the overall clarity about the different systems.

The review can be accepted after a minor revision.

Author Response

The review article “Liver organoids as an in vitro model to study primary liver cancer” by Siervi and Turato discusses recent findings related to the development and importance of Liver organoids.

It’s an interesting and translationally relevant for liver cancer. The review provides a comprehensive understanding about the organoids as well as discusses pros and cons of different model system. Only one minor comment needs to be improved for the betterment of the manuscript. On page 4, making subsections for section 3 for topics like spheroid, scaffold-based systems, 3D-bioprinting, and organs-on-a-chip will improve the overall clarity about the different systems.

Author’s response to the reviewer 2

We would like to thank you for your valuable comment about the manuscript. As you suggest, we provided to make subsections for spheroids; scaffold-based systems; and 3D-bioprinting and organ-on-a-chip.

Round 2

Reviewer 1 Report

The authors have answered each question though further justified clarification in the manuscript would have been better. From my side, it can be accepted.